# Biomolecular Basis of Life

**DOI:** 10.3390/metabo15060404

**Published:** 2025-06-16

**Authors:** Janusz Wiesław Błaszczyk

**Affiliations:** Jerzy Kukuczka Academy of Physical Education, 40-065 Katowice, Poland; januszwblaszczyk@gmail.com

**Keywords:** life, energy metabolism, information metabolism, cellular turnover, senescence, death

## Abstract

Life is defined descriptively by the capacity for metabolism, homeostasis, self-organization, growth, adaptation, information metabolism, and reproduction. All these are achieved by a set of self-organizing and self-sustaining processes, among which energy and information metabolism play a dominant role. The energy metabolism of the human body is based on glucose and lipid metabolism. All energy-dependent life processes are controlled by phosphate and calcium signaling. To maintain the optimal levels of energy metabolism, cells, tissues, and the nervous system communicate mutually, and as a result of this signaling, metabolism emerges with self-awareness, which allows for conscience social interactions, which are the most significant determinants of human life. Consequently, the brain representation of our body and the egocentric representation of the environment are built. The last determinant of life optimization is the limited life/death cycle, which exhibits the same pattern at cellular and social levels. This narrative review is my first attempt to systematize our knowledge of life phenomena. Due to the extreme magnitude of this challenge, in the current article, I tried to summarize the current knowledge about fundamental life processes, i.e., energy and information metabolism, and, thus, initiate a broader discussion about the life and future of our species.

## 1. Phenomenon of Life

Life is a self-sustaining process that takes place in highly organized organic systems, characterized by being preprogrammed, interactive, and susceptible to epigenetic and evolutionary adaptation [1]. In realizing life processes, multiple metabolites play fundamental functions such as energy fuel, structural and catalytic substrates, and signaling particles, enabling systemic synergy and environmental interactions. Self-awareness, self-sustaining, and motor activity are other attributes of individual life [1]. Life is considered a characteristic of the biological system that preserves, furthers, or reinforces its existence in a given environment. It is defined descriptively by the capacity for metabolism, homeostasis, self-organization, growth, adaptation, response to stimuli, and reproduction. A living organism forms a thermodynamic open system with an organized molecular structure that can evolve and reproduce. To survive in a changing environment, the body must efficiently control homeostasis, providing a stable physicochemical environment for the proper course of spontaneous biochemical reactions at the cellular, tissue, and systemic levels. Energy metabolism plays a pivotal role here, which relies on energy fuels such as glucose, fatty acids, and oxygen. The optimal functioning of the organism depends on stable body temperature, fluid balance, and pH. Moreover, homeostasis is maintained by signaling ions (sodium, potassium, calcium, and phosphate), lipids, and hormones. All these must be precisely controlled despite changes in environmental conditions, diet, and motor activity. From a thermodynamics perspective, living organisms must maintain thermal, mechanical, and chemical equilibria to ensure the proper course of biochemical reactions essential to life.

Life processes, regardless of the complexity of the system, are carried out according to the same algorithms, the aim of which is to ensure constant conditions for the course of metabolic processes. The condition of life is to maintain homeostasis at all levels, from cellular, through tissue, to systemic. Regardless of dietary habits or environmental changes, metabolic processes must be carried out in relatively stable physicochemical conditions that allow for the efficient implementation of life functions. The algorithm of all life processes, regardless of the level of implementation, has a similar scheme. Energy metabolism ensures homeostasis and signal (information) metabolism triggers functional processes, confirms the performance of a given function, and returns the cell (or system) to a resting state. The energy and functional capabilities of the organism are limited, and therefore all elements of life processes must be carried out with the greatest efficiency. This applies to the acquisition of metabolic substrates, their use and recycling, and the removal of waste products. Any violation of the optimization principle can impair the implementation of life processes, leading to the wear and aging of cells.

Life depends on many precisely controlled and coordinated, energy-dependent biochemical processes. Of particular note is the fact that this control is multimodal and multilevel, reaching from the subcellular level to interpersonal and environmental relationships. Firstly, life is a complex form of energy exchange, thus, energy metabolism must be considered as the foundation of life. The body’s energy balance is achieved by the interaction of several vital processes such as respiration, the acquisition and distribution of energy metabolites, and removing of the waste product. The uninterrupted respiratory and circulatory activity provides the living organism with oxygen, glucose, and fatty acids, which are essential for life at the cellular level [2,3]. Consequently, monitoring and control of these metabolites in the blood is a source of life signaling for all tissues, particularly for the brain. The brain, based on these signals, coordinates internal interorgan interactions and also controls organismal behavior. This behavioral control concerns predominantly all aspects and forms of motor activity, including locomotion, physical work, respiration, and blood circulation, which are the main energy-consuming contributors of energy consumption. In addition, the digestive processes dependent on smooth muscle activity may impact significantly energy balance. Importantly, all muscular activities are performed under the control of calcium-phosphate metabolisms and signaling [4]. Due to limited energy resources, the implementation of processes with high energy consumption is spread over time in the circadian cycle. Alternating states of wakefulness and sleep allow for the temporal separation of energy-consuming processes requiring conscious nervous control (such as motor activity, work, food consumption, etc.) from regenerative, developmental, and repair processes that also require high energy expenditure, but can be carried out without the participation of consciousness.

## 2. Algorithm of Life

All life processes are carried out according to a fixed algorithm (Figure 1). They begin with the acquisition of energy substrates, especially oxygen and energy fuels such as glucose and fatty acids. These substrates are then distributed to all cells of the body to maintain their energy metabolism. In addition, excess glucose and fatty acids are stored for use in intermealperiods. Cellular life processes also include specific secretory functions. All these functions are sequences of biochemical processes that culminate in the restoration of homeostasis by recycling substrates and removing waste products.

Foods consumed by humans contain several energy components, of which glucose and fatty acids constitute the basis of the body’s energy metabolism [2]. In response to the postprandial rise in blood glucose levels, the pancreas-released insulin initiates glycogenesis in skeletal muscles, liver, and adipose tissue. In humans, the majority of glycogen is stored in skeletal muscles (up to 500 g) and the liver (100 g) [5,6,7]. Importantly, almost 80% of the glycogen is stored in skeletal muscles, which are the tissues that transform chemical energy into mechanical work and, therefore, uses most of the glucose-derived energy during motor activity. In contrast to autonomic muscular stores, the liver glycogen complements blood glucose levels, to maintain the energy supply to the brain and other glucose-consuming tissues. Maintenance of blood glucose concentration in a narrow range is the major physiological priority [6,7]. Four grams of glucose circulate in the blood, and the brain requires approximately 60% of it for its functioning [6].

Two functional states, differing fundamentally in energy metabolism, can be distinguished in the control of life processes. The base is the resting state, which aims to maintain cellular metabolic processes at a level necessary for maintaining homeostasis, with minimal energy consumption. The active state, on the other hand, is characterized by a selective increase in metabolic processes in tissues performing a given life activity, e.g., in skeletal muscles during physical activity. Limited energy resources stored in muscle tissue cause an increase in the conversion of biochemical energy into physical work, which is realized thanks to the trophic/functional synergy, i.e., increased blood supply to the muscles performing work while limiting blood flow in other muscles and tissues.

Resting and action-related energy metabolisms shape the energy balance of all living organisms. The first is characterized by low energy consumption while focusing on maintaining the homeostasis necessary for the optimal functioning of all body organs. In the resting state, even the continuously acting muscles, including the heart and diaphragm, and secondary respiratory muscles, work with a slower rhythm, thus minimizing energy consumption. The skeletal muscles in the resting state, despite their large mass, do not use significant amounts of energy, and their activity is limited to the storage of glucose fuel and maintaining cellular homeostasis. The crucial and energy-consuming element of homeostasis is the clearance of waste products generated during a state of activity. Focusing on energy metabolism, one can claim that the brain’s motivational system, depending on the energetic status of the entire organism, regulates the rest/activity rhythm. The action state is characterized by the elevated activity of the neuromuscular system. Although the transformation of biochemical energy into mechanical work consists of a supreme load for the neuromusculoskeletal system, it usually requires the synchronized activity of several organs, such as the liver, kidneys, and circulatory and respiratory systems.

The heart is the primary pacemaker of life. It is equipped with a sinoatrial node that spontaneously generates electrical impulses, which initiate each heartbeat. Myocardium physiology and its energy metabolism depend primarily on fatty acid oxidation, only partially supported by glucose metabolism [8]. The albumin-bound free fatty acids and lipoproteins, both chylomicrons and very low-density lipoproteins, are the primary energy fuel for the heart (up to 70% of energy) [8]. The dependence of cardiac energy metabolism on lipids is seen in the specific energy fuel supply. Chyle, formed in the small intestine during fat digestion, is supplied by the lymphatic thoracic duct directly to the heart to fulfill its metabolic needs. The lymph circulation is forced by the respiratory action of the diaphragm, aided by the duct’s smooth muscle and internal valves, which prevent the lymph from flowing back down again [9]. The excess of chyle gets into the blood and undergoes liver filtration. To meet myocardium metabolic demands, the fatty acid processes must be accompanied by efficient glucose and lactate metabolism that supply the protons necessary for mitochondrial respiration and massive ATP production [8]. Increased myocardial lipid deposition occurs when fatty acid uptake by cardiomyocytes exceeds their oxidation, resulting in metabolic dysfunctions such as liver steatosis, insulin resistance, and lipotoxicity. The excessive accumulation of lipids and over-activation of lipid signaling pathways trigger cellular distress and dysfunction, which may manifest as insulin resistance, defective mitochondria, energy starvation, and endoplasmic reticulum stress and may ultimately lead to apoptotic cell death and heart lipotoxic cardiomyopathy [8].

The circadian cycle allows the most energy-consuming processes to spread over time. During the day, processes related to motor activity dominate, and their implementation requires conscious nervous control. Energy used by muscles is undoubtedly the most important contributor to the body’s energy balance. In the neuromuscular system, the largest energy expenditures are primarily related to processing biochemical energy into muscle work. The circadian regulation of hematopoietic stem cells release into circulation is coordinated by the extracellular ATP-dependent pathway [10,11].

## 3. Energy Metabolism as a Basis of Life

Cellular respiration is a set of metabolic reactions and processes that convert the chemical energy of glucose and oxygen to drive the bulk production of NADH and adenosine triphosphate. NAD^+^ is an essential pyridine nucleotide that serves as a rate-limiting cofactor and substrate for critical cellular processes involved in oxidative phosphorylation [12]. In the reduced form, NADH serves as a hydride donor in ATP synthesis in mitochondria. Some NAD^+^ is also converted into the coenzyme nicotinamide adenine dinucleotide phosphate (NADP), participating in antioxidant defense and regulating cellular signaling via NADPH oxidases.

At the cellular level, NAD^+^ is continually turned over by three classes of NAD^+^-consuming enzymes: CD38/glycohydrolase, the protein deacetylase family of sirtuins, and PARPs, which have various important cellular functions impacting metabolism, genomic stability, gene expression, inflammation, circadian rhythm, and stress resistance [12,13,14,15]. Sirtuins (SIRT) regulate energy homeostasis by controlling the acetylation status and activity of several enzymes and transcriptional regulators. SIRT3 is the major mitochondrial deacetylase serving as an important regulator of energy metabolism and is highly expressed in metabolically active tissues such as the brain, skeletal muscle, liver, kidney, heart, and adipose tissue [16].

In young, healthy individuals, high NAD^+^ levels ensure the proper conduct of all biochemical processes necessary for cell life. NAD^+^ deficiency due to increasing demands for NAD-consuming enzymes, as occurs in aging, causes the mitochondria to become less efficient and produce less ATP [17]. Sirtuin activity in older adults might be compromised due to the systemic decline in NAD^+^ [18]. Cellular NAD^+^ levels decline steadily to reach 100% of the brain’s physiological needs by the age of 45 [19]. In old age, the NAD^+^ deficit can reach up to 20% of physiological needs, leading to massive death of neurons and disintegration of their neuronal networks. The decline in SIRT1 activity downregulates mitochondrial biogenesis, oxidative metabolism, and antioxidant defense pathways, leading to damage to complex I of the electron transport chain. Oxidative stress increases the amount of incorrectly formed proteins that cannot be fixed due to a decrease in the synthesis of NAD-dependent enzymes. As a consequence, the cellular energy deficit and metabolic crisis are rapidly escalating. When the energy deficit exceeds a critical level, oxidative phosphorylation arrests cellular respiration and initiates the process of programmed death. Activation of the nuclear enzyme, poly(ADP-ribose) polymerase-1 (PARP-1), plays a critical role in various cellular responses to injuries, mainly as a death messenger [17].

The deficit of NAD^+^ impairs cellular energy metabolism, thus escalating cell senescence. NAD^+^ deficiency causes failure of the cellular respiration process and the breakdown of all cellular NAD-dependent repair processes, causing massive cell death. This process can no longer be compensated for by cellular turnover. Importantly, in young healthy individuals, the cellular level of NAD^+^ is efficiently supplemented by tryptophan. Still, this process breaks down due to chronic stress associated with massive cell death and metabolic diseases. The kynureninic pathway produces quinolinic acid instead of NAD^+^, which leads to the breakdown of information metabolism and the loss of the motivational and emotional drive necessary for life.

Cellular life strongly depends on the membrane’s ability to precisely control the exchange of solutes between the internal and external compartments [20]. Choline is a nutrient essential for life and needed for the structural integrity and signaling functions of cell membranes, normal cholinergic neurotransmission, normal muscle function, and lipid transport from the liver [20,21]. It is also critical for cell proliferation and apoptosis.

The glucose–fatty acid synergy is a biochemical mechanism that controls fuel selection and adapts substrate supply and demand in normal tissues in coordination with hormones controlling substrate concentrations in the circulation [22]. The hormones controlling lipolysis affect circulating concentrations of fatty acids, which in turn control glucose metabolism in the liver and skeletal muscles [22]. Adipose tissue lipolysis is inhibited by glucose and insulin. The proportions between glucose and lipid metabolism are determined in part by the different contributions of metabolic processes by individual organs and cells. In the heart and kidney, β-oxidation of fatty acids provides more than 90% of ATP with a simultaneous reduction in the NADH-to-NAD^+^ ratio [23]. The mitochondria can oxidize fatty acids at the highest rate only with fully functional respiratory complex I when pyruvate and succinate are present [23]. In these conditions, mitochondria in the heart and kidney maintain high rates of respiration and ATP production even at the maximal functional loads in these organs [23].

Lipids are essential for brain functioning, participating in synaptogenesis (learning and memory) as well as repair and cellular turnover. Sphingolipids are essential constituents of all eukaryotic membranes. The sphingolipid signaling mediates processes involved in cellular turnover, inflammation, and programmed cell death [24]. Therefore, the control and setting of physiological proportions between glucose and lipid metabolism are fundamental for health and life. In highly active cells like neurons and myocytes, ionic channels coordinate and control several life processes such as neurotransmission, secretion, muscle contraction, growth, proliferation, and migration [25]. Ionic channels allow for the controlled transmembrane transport of Na^+^, K^+^, Cl^−^, and calcium ions [20,25]. Fat accumulation in the extracellular matrix, however, impairs the functioning of the ionic channels, making cellular signaling dysfunctional [25]. Chronic increase in concentration of free fatty acids leads to lipotoxicity, mitochondrial dysfunction, and insulin resistance. The 40% decrease in inner membrane potential may abolish ATP synthesis completely, which opens a vicious circle of cellular senescence and death.

### 3.1. ATP Function in Energy Metabolism

In living systems, all bioenergetic processes are coupled via adenosine nucleotides [26,27]. The most important regulatory elements involved in the coupling of catabolic and anabolic reactions are ATP, ADP, and AMP. The adenosine nucleotides are not only tied to the metabolic pathways involved in the cell’s energetic system but also act as allosteric control of numerous regulatory enzymes, and allowing changes in adenosine nucleotide levels can practically regulate the functional activity of the overall multi-enzymatic network of the cell. The heart, kidney, and diaphragm are the main sources of ATP entering the bloodstream in the resting phase [28,29]. Mitochondria in cardiomyocytes constitute approximately one-third of the cell volume and produce more than 95% of the ATP in the myocardium [29]. To maintain the optimal life-sustaining level of ATP production, the heart permanently consumes large quantities of energy, and fatty acids seem the preferred substrate, accounting for 60–90% of the myocardium’s energy supply [30]. It has been estimated that cells within the human body depend upon the hydrolysis of 100 to 150 moles of ATP per day to ensure proper functioning and life [31].

In energy metabolism, ATP is the primary life signaling molecule that coordinates all life processes at cellular, tissue, and systemic levels. At rest, heart-derived ATP released to the circulation is spontaneously hydrolyzed, while the kinetics of this process are regulated by magnesium ions. Entering the circulation, ATP, if not combined with magnesium ions, is unstable and prone to irreversible hydrolysis, maintaining heat production and elevating the concentration of inorganic phosphate [32]. The phosphocreatine system can also produce high levels of ATP in situations of high metabolic demand, such as during the initiation of high-intensity motor activity when the rate of ATP use exceeds its capacity for generation by other metabolic pathways [33,34]. The phosphocreatine serves as a rapidly mobilizable reserve of high-energy phosphates in skeletal muscle, myocardium, and the brain.

The ATP-dependent potassium channels (KATP) play a vital role in energy metabolism adjustment in all insulin target tissues, including the brain, heart, liver, kidneys, and skeletal muscles [4,25,35]. The KATP channels regulate physiological processes, including hormone secretion, vascular tone, learning and memory, and cardiac and neuronal protection against ischemic insults [35]. They are responsible for the endothelial cells and pericytes’ functioning and interaction. In particular, endothelial cells regulate exchanges between the bloodstream and the surrounding tissues. Pericytes, in turn, help in the maintenance of homeostatic and hemostatic functions of the central nervous system. In the brain, pericytes regulate capillary blood flow and the clearance and phagocytosis of cellular debris. In the pancreas, KATP channels are primarily responsible for maintaining the β-cell membrane potential and are involved in depolarization-mediated, calcium-dependent insulin release [25].

The ATP entering the bloodstream acts as a trophic signaling molecule and as a transmitter or a co-transmitter in the peripheral and central nervous systems. The extracellular ATP, however, can be rapidly hydrolyzed as a short-term neurotransmitter in neuromodulation, secretion, and neurotransmission, and plays a role as a long-term signaling molecule in cell differentiation, proliferation, and death [36,37]. Thus, the activation of purinergic receptors contributes mostly to stem cell differentiation and neuronal function [38]. Importantly, the circadian regulation of hematopoietic stem/progenitor cells release into peripheral blood is coordinated by the extracellular ATP-associated pathway [10,11].

The concentration ratio of ATP to ADP is known as the “energy charge” of the cell. The uptake of cytosolic ATP into the endoplasmic reticulum lumen is critical for the proper functioning of chaperone proteins that assist the conformational folding or unfolding of large proteins or macromolecular protein complexes [39]. The endoplasmic reticulum uses energy from ATP hydrolysis for protein folding and trafficking [40]. Under physiological conditions, increases in cytosolic Ca^2+^ inhibit mitochondrial ATP production. Consequently, the endoplasmic reticulum ATP level is readily depleted by inhibition of oxidative phosphorylation, leading to protein misfolding [40].

### 3.2. Role of Insulin and Insulin-like Growth Factor in Energy Metabolism

Glucose, fatty acids, and amino acids are the main substrates an organism can use to maintain metabolic homeostasis [41]. Fatty acids are also building blocks for the biosynthesis of molecules [41]. Under physiological conditions, glucose is the preferred substrate for energy metabolism in the neuromuscular system, while during fasting, fatty acids and ketone bodies may become a dominant energy source [22,41]. Fatty acid oxidation is exceptional in the heart, whose energy metabolism relies on fatty acids for a major proportion (60–90%) of its energy needs [22].

The sense of taste plays a special role in establishing food preferences necessary for maintaining life. The sweet taste receptor is involved in nutrient sensing, monitoring changes in energy storage, triggering metabolic and behavioral responses, and maintaining the energy balance [42,43]. Sugar preference and intake are controlled at least on three levels: gustation, gut–brain axis, and brain-glucose sensing [42]. The brain can sense glucose via oral and visceral sensations. Nutrient sensing initially occurs in the gastrointestinal tract. The gut sends signals to the rest of the body, including the brain, about current nutritional status by secreting hormones, such as ghrelin, gastric inhibitory peptide, peptide YY, cholecystokinin, glucagon-like peptide-1, and serotonin that are important regulators of glucose and energy homeostasis [44].

In young healthy individuals, insulin secretion is tightly controlled to maintain steady plasma glucose levels despite its continuous use by all tissues. The glucose control is based on uninterrupted insulin and glucagon secretion patterns complemented by markedly increased prandial insulin secretion [45,46]. The estimated biological half-life of insulin in the bloodstream lies between 3 and 10 min. Insulin activity in the liver is highly downregulated by free fatty acids, reaching even the threshold of insulin resistance.

The insulin-like growth factor (IGF-1) is a protein with high sequence similarity to insulin. It mediates anabolic biological processes, including the increase in glucose metabolism, glycogenesis, lipid and protein synthesis, and the inhibition of gluconeogenesis, lipolysis, and protein degradation. Its action is mainly dedicated to energy metabolism during body development. Insulin and IGF-1 share the same tyrosine kinase receptor; therefore, IGF-1 complements the function of insulin [47]. However, they differ substantially in their half-life, and IGF-1 is a relatively stable protein produced during sleep by the liver in response to growth hormone release.

IGF-1 exerts anabolic action on the skeleton by modulating the effects of parathyroid hormone and through crosstalk with mechanosensory pathways. IGF-1 responsiveness in osteocytes and osteoblasts is increased after mechanical load, whereas the parathyroid hormone promotes the proliferation and maturation of osteoblast precursors and exerts anti-apoptotic activity on osteoblasts [47]. IGF-1 is also known to induce muscle cell proliferation and hypertrophy by activating the PI3K/Akt/mTOR signaling pathway [47]. In the brain, IGF-1 regulates neural development, neurogenesis, myelination, dendritic branching, and synaptogenesis [47]. Receptors for IGF-1 are found in vascular smooth muscle. The systemic production of growth hormone and IGF-1 declines, steadily reaching very low levels in humans aged 60 years [47,48]. In adults, alterations in the activity of the growth hormone and IGF-1 have been documented in aging, stress, hormonal changes, motor activity, nutrition level, disease state, and circadian rhythm [47].

### 3.3. Cellular Turnover and Its Role in Tissue Repair

During ontogenesis, the cellular composition of the organism is precisely tailored to its behavioral aims and needs as well as the availability of particular metabolites and substrates in a given ecosystem. The body’s anthropometry and metabolism mostly match the proper course of energy processes. Therefore, maintaining optimal endogenous synergy is a primary condition for an organism’s healthy and physiological functioning. For this reason, all inefficient, damaged, or dysfunctional cells must be immediately and effectively replaced by resident progenitor cells. Apoptosis, necroptosis, and pyroptosis are genetically programmed cell death mechanisms that eliminate obsolete, damaged, infected, and self-reactive cells [49]. Apoptosis is associated with mechanisms that minimize the immune response, whereas necroptosis and pyroptosis release proinflammatory molecules that amplify the immune response to noxious stimuli, such as infections. Only the long-lived postmitotic cells, such as cardiomyocytes and neurons, are excluded from cellular turnover. Their viability has a decisive impact on individual health and lifespan [50].

Cellular turnover plays an important role in the adaptation processes [50]. In this process, cells are exchanged in individual tissues and organs, with the main goal of cellular exchange being to achieve the most efficient and balanced cellular activity. This highly energy-consuming process occurs at different rates in individual tissues, with the cells of the intestinal lining being the fastest to be replaced, and the largest population of cells being replaced are erythrocytes [50]. In contrast, osteocytes, characterized by long lifespans, constitute 90–95% of all bone cells in the adult skeleton [51]. Bones are constantly renewed through a mechanism called bone remodeling [52]. The skeleton remodeling cycle takes approximately 120 days, and 10% of the human skeleton is normally remodeled every year [53]. The process requires the differentiation of mesenchymal stem-cell-derived osteoblasts with subsequent collagen synthesis. While osteoblasts and osteoclasts are viable from days to a few weeks, the osteocytes can live in the bone matrix for decades [54].

The human musculoskeletal system forms an interface between the organism and the environment. The gravitational force plays an important role in the development of the musculoskeletal system and its anatomical adaptation [52,55,56]. In response to mechanoelectric stimulation, osteoblasts secrete a large matrix volume to become osteocytes anchored in it [56]. Mechanotransduction by Piezo channels regulates osteoblast/osteocyte activity and, thus, strengthens the skeleton, enabling it to adapt to a wide range of mechanical loadings [57,58]. Piezo channels play essential roles in the development of the cardiovascular system, lungs, cartilage, and bones. They serve as mechanosensors for touch, pain, proprioception, hearing, vascular tone, and fluid flow. Piezo signaling in gut epithelia controls gut peristalsis and regulates bone density by serotonin production [58].

The debris of the worn-out and apoptotic cells must be efficiently degraded, since their accumulation may compromise cellular turnover and functioning. Particularly, long-lived postmitotic cells, such as cardiac myocytes, neurons, and retinal pigment epithelial cells, progressively accumulate harmful waste products [59]. Lipofuscin deposition hampers autophagic mitochondrial turnover, promoting the accumulation of senescent mitochondria, which are deficient in ATP production but produce increased amounts of reactive oxygen species. Increased oxidative stress and damage to mitochondria and lysosomes impair cellular energy metabolism and homeostasis, culminating in cell death [59]. This vicious circle strikes at the basis of cellular homeostasis by changing substrate concentration, cytoplasmic/lysosomal pH, and temperature. It also impairs the other determinants of life, i.e., the signaling metabolism.

The mitochondria are the cellular hub that converts energy fuels into life processes. In the realization of this task, mitochondria must interact and exchange materials with other cellular organelles, such as the endoplasmic reticulum and the lysosomes [60]. Deterioration in mitochondrial function is one of the twelve currently established hallmarks of aging [61]. Mitochondria are dynamic cellular organelles that are the major hub for converting energy for cellular processes. Dysfunction of mitochondria is the key pathomechanism of cellular aging and death. Long-lived postmitotic cells show dramatic age-related alterations affecting mainly mitochondria and the lysosomal compartment. Mitochondria are primary sites of reactive oxygen species formation that cause progressive damage to mitochondrial DNA and proteins in parallel to intralysosomal lipofuscin accumulation [62,63].

## 4. Central Control of Energy Metabolism

The brain is the main controller of life; thus, its proper function is the main determinant of organismal integrity. The brain monitors oxygen and glucose circulating levels, which are the foundations of its energy metabolism and functioning [5,64,65]. Blood glucose and oxygen levels control the motivational/emotional status of the organism by the release of catecholamines, which are the primary activators of lipolysis. Also, other hormones, including cortisol, glucagon, growth hormone, and adrenocorticotropic hormone, have similar effects. Systemic glucose decline activates hormonal signaling that induces compensatory lipolysis by stimulating hormone-sensitive lipase activity in the adipose tissue [22]. Consequently, lipids become an important participant in energy metabolism in case of stressful or conflict situations, the resolution of which requires additional energy. Blood glucose level is a sensitive indicator of energy balance used by the brain to control the circadian cycle. It is responsible for the sleep/wake transition based on the catecholamine-dependent release of calcium and phosphate ions from the bones. Thus, it constitutes a metabolic trigger allowing the body to switch from a resting to an active state.

The hypothalamus is a brain region that plays a critical role in the regulation of energy homeostasis [66]. It is the central controller of systemic metabolic processes including body temperature, hunger, thirst, fatigue, sleep, and circadian rhythms. The hypothalamus is a key regulator of metabolism, controlling resting metabolism, activity levels, and responses to external temperature and food intake [66]. Supervision by the brain of activities necessary to maintain the body’s life is another process characterized by high energy consumption [67]. The sleep phase, in turn, is characterized by a lack of motor activity and the exclusion of conscious and motivational-emotional processes. However, only a small decrease in the brain’s energy consumption is observed during sleep [68]. This phenomenon is primarily associated with the activity of the processes of restoring brain homeostasis during sleep, including the removal of waste products of nervous activity. In addition, repair processes and memory-forming processes are carried out in the sleep phase [68]. The restructuring of synaptic connections of neural networks and the process of neurogenesis in GABAergic subcortical structures of motor memory require significant energy expenditure [65,69]. The processes of neuronal homeostasis recovery, restoring ionic balance, renewing neurotransmitter resources, and the removal of waste products consume a significant amount of energy, stored in the liver in the form of glycogen and fatty acids. The substantial decline in hepatic glucose resources during sleep is considered energy stress that triggers the sleep–wake transition and restores energy resources by feeding behavior. The transition is controlled by the release of catecholamines [70].

Perception, learning, and memory are important attributes of life. These processes are modulated by emotions affecting and motivating all aspects of human behaviors and supporting survival. Emotional states coordinate also food habits, which are the basis of energy metabolism and homeostasis. The appetitive states usually involve reward signaling including dopamine, serotonin, and oxytocin [71]. On the other hand, major metabolic deficiency is signaled by fear and even panic. The posterior parietal cortex is the area important for sensory-motor integration [72]. Among its functions is the forming of intentions, that is, high-level cognitive plans for movement [72].

Acetylcholine is the primary neurotransmitter of the parasympathetic nervous system. The cholinergic system is a branch of the autonomic nervous system that plays an important role in memory, digestion, heartbeat control, blood pressure, and movement [21]. The brain contains several cholinergic areas, each with distinct functions, such as arousal, attention, memory, and motivation [73].

Arousal is the physiological state of being awoken or of sense organs stimulated to a point of perception. It involves activating the ascending reticular activating system in the brain, leading to increased heart rate, blood pressure sensory alertness, mobility, and reactivity. Low levels of norepinephrine allow one to sleep and an increase initiates wakefulness. ATP acts as either the sole transmitter or a co-transmitter in most nerves in the peripheral and central nervous system [37]. Noradrenaline and ATP are sympathetic co-transmitters. In the pancreas, there is an increased release of glucagon. In the liver, there is an increase in the production of glucose, either by glycogenolysis or by gluconeogenesis. In skeletal muscles, there is an increase in glucose uptake. The central effects of noradrenaline are manifested in alertness, arousal, and readiness for action.

The vagus nerve innervates numerous organs including the gastrointestinal tract [74]. The vagus is a mixed nerve, with 80% afferent fibers that convey visceral, somatic, and taste signals. The rest are efferent fibers controlling gastrointestinal motility and secretion as well as blood circulation. The tissue signals via vagus nerve afferents are sent directly to sympathetic neurons in hindbrain nuclei, which project to the hypothalamus [44] and to the limbic structures, which affect and motivate all aspects of life [73].

The sympathetic nervous system controls the body’s vital organs and functions, such as the cardiovascular and respiratory systems. It contributes to the stress response by releasing epinephrine and norepinephrine from the adrenal medulla. Sympathetic tone may mediate the leptin-dependent regulation of bone-related calcium metabolism [70]. An elevated sympathetic tone accompanies stress related to between-meal glucose decline that motivates organisms to consummatory behaviors to replenish energy fuels and glucose, in particular. The behavioral response is triggered by bone calcium release controlled by catecholamines [70].

The bone-derived hormone osteocalcin crosses the blood–brain barrier and binds specifically to serotonergic neurons of the raphe nuclei in the brainstem, to neurons of the CA3 region of the hippocampus, and of the dopaminergic nucleus of the ventral tegmental area in the midbrain [75]. In the brain, osteocalcin plays an important role in development and functioning including spatial learning and memory [76]. The osteocalcin deficiency could be traced to a decrease in the synthesis of all monoamine neurotransmitters and to an increase in GABA [75]. This is accompanied by behavioral phenotypes such as increased anxiety and a profound deficit in spatial learning and memory [75,76]. Osteocalcin permits manifestations of the acute stress response by inhibiting the post-synaptic parasympathetic neurons, thereby leaving the sympathetic tone unopposed.

### 4.1. Calcium Signaling in the Neuromuscular System

Calcium entering the neural and muscle cells through voltage-gated calcium channels serves as the second messenger of electrical signaling, initiating multiple cellular events [77]. In motoneurons, voltage-gated calcium channels trigger the release of acetylcholine neurotransmitters [77]. In the myocardium, activation of calcium channels initiates contraction directly by increasing cytosolic Ca^2+^ concentration and indirectly by activating calcium-dependent calcium release by ryanodine-sensitive Ca^2+^ channels. In skeletal muscle cells, voltage-gated Ca^2+^ channels in the transverse tubule membranes interact directly with ryanodine-sensitive Ca^2+^ release channels and activate them to initiate muscle contractions, enabling locomotory movements [77].

### 4.2. Sialic Acid

Sialic acid is an essential component of brain gangliosides and sialylated glycoproteins, particularly as precursors for the synthesis of the polysialic acid (polySia) glycan that post-translationally modifies the cell membrane-associated neural cell adhesion molecules (NCAM) [78]. Polysialylated NCAM and neural gangliosides play critical roles in mediating cell-to-cell interactions involved in neuronal outgrowth, synaptic connectivity, and memory formation. The polySia moiety on NCAM confers unique properties that influence many cellular processes, including cell migration, neurite outgrowth, branching, neuronal pathfinding, regeneration, and synaptic plasticity. In humans, the predominant form of sialic acid is N-acetylneuraminic acid (Neu5Ac). This is the key monomeric precursor of brain glycoproteins, including polySia (polySia), gangliosides, glycosaminoglycans, and mucins. These sialoglycoconjugates are ubiquitously expressed throughout the human body.

Sialic acid consists of negatively charged sugars that occupy terminal positions of oligosaccharide chains of most glycans (glycoproteins and gangliosides) [79]. Free sialic acid salvage from the degradation of recycled glycans occurs in lysosomes, and free sialic acid exits lysosomes into the cytosol through the SLC17A5 membrane transporter. The proton-driven transporter SLC17A5 plays a role in determining lysosomal pH. Changes in the intra-lysosomal milieu due to SLC17A5 deficiency, resulting from reduced trafficking of protons or acidic sugars, may affect other lysosomal functions. Defective SLC17A5 leads to intra-lysosomal free sialic acid accumulation with progressive neurodegenerative symptoms including muscular hypotonia, cerebellar ataxia, and cognitive impairment. Defective SLC17A5 leads to intra-lysosomal free sialic acid accumulation with progressive neurodegenerative symptoms including muscular hypotonia, cerebellar ataxia, and cognitive impairment.

Sialic acids are commonly found on mammalian cell surfaces as the final component of glycoconjugates and play important roles in cellular recognition, proliferation, differentiation, and motility. The glycocalyx is a layer of glycoproteins and glycolipids that surrounds cells. It protects them from damage and external factors such as dehydration and toxins. The glycocalyx mediates cell-to-cell signaling, which is important in vital processes such as development, tissue physiology, repair, and immune response. The glycocalyx of the intestinal epithelium is essential for the absorption of nutrients. The endothelial glycocalyx is particularly important in regulating vascular permeability and can transmit extracellular mechanical stimuli, influencing vascular permeability, tension, inflammation, and signaling.

The complexity, interplay, and differentiated kinetics of particular biochemical reactions require that particular phases of the biochemical processes should be well-temporary and quantitively coordinated. Therefore, homeostasis and life depend on uninterrupted intracellular and between cells, and tissue, communication. The signaling metabolism being the foundation of life includes self-organizing synergies that optimize cellular and tissue functioning and interactions, which culminate in cognitive and social processes [80]. The brain is the master controller of energetic and information metabolisms [66]. Communications and interactions at the cellular and organ levels provide feedback for the brain which may have a decisive impact on emotional/motivational states, health, and longevity of individuals.

Several organismal capabilities such as motivations, emotions, perception, learning, and memory allow humans to adapt socially and maintain homeostasis for a lifetime in our ecosystem. In particular, emotions coordinate the homeostasis of an organism in a complex, dynamic environment and participate in the regulation of social behaviors [81]. Simultaneously, the endogenic homeostatic controls including energy metabolism and intercellular and interorgan signaling and interactions allow for maintaining organismal integrity and life. In these processes, stress-dependent regulation of behavior plays a vital role. Catecholaminergic, dopaminergic, and serotonergic systems contribute to the control of vital life processes such as motor activity, cognition, emotion, and memory [82]. They also co-regulate the brain and immune system interactions. Serotonin and dopamine are neurotransmitters related to fatigue, a feeling that adjusts intensity or interrupts skeletal muscle activity [83]. All changes in peripheral physiological systems such as substrate depletion or metabolite accumulation act as afferent signals, which modulate control processes in the brain [83].

The most important product of information metabolism is self-awareness [84]. Perception and memory enable our brain to form a somatosensory representation of our organism and an egocentric representation of the environment [85,86]. These brain networks of consciousness allow us to perceive the existence of ourselves in the environment, which is the basis for motivational and emotional control of our behaviors necessary to survive. The hippocampus is the main brain structure that integrates interoceptive information with sensory signals allowing to communicate the body with the environment [87]. At the basic level, communication determines the affective behavior of the organism in the exploration of the environment to obtain food, as well as avoiding life-threatening situations through ‘flight or fight’ responses, and procreation [87]. The next aim of information metabolism is the formation of social attachments which is a critical component of human life. In the brain, vasopressin and oxytocin neuropeptide systems are critical for the establishment of social bonds and the control of emotional behaviors. In humans, these processes begin early in postnatal life and play a critical role in children’s survival and environmental adaptation. Therefore, both neuropeptide systems are associated with social and emotional communication, which are the pillars of information metabolism. They allow us to form emotional bonds and, what is also important, reduce stress [88].

The base of information metabolism is multilevel and multimodal communication within organisms as well as with the environment. It allows for maintaining life synergy through integrated endocrine, paracrine, and autocrine signaling at all levels of the organism, from cellular to systemic. Mitochondria are vital in cellular and organismal pathways that direct metabolism, stress responses, immunity, and cellular fate [89]. Towards this aim, mitochondria have established networks of both intra- and extracellular communication. Intracellular communication routes comprise direct contacts between mitochondria and other cellular components using ions, metabolites, and other intracellular messengers [89]. Mitochondrial cytokine (mitokine) factors can provide between-tissue communication and may respond to immune signaling from extracellular sources [89]. Mitochondrial signaling includes the transport of metabolites or small molecule messengers as well as the budding of vesicles carrying mitochondrial cargo to other cellular locations [89].

The mitochondrial release of cytochrome c is pivotal for the control of apoptosis [89]. The physical contact sites with other organelles and subcellular compartments allow for mitochondria-driven transcriptional responses to rewire cellular metabolism in response to stress [89]. The mitochondrial-to-nuclear transcriptional programs direct the activity of nutrient sensors such as mTORC1 and AMPK and protein homeostasis machinery such as lysosomes, as well as the proteasome, chaperones, and small heat shock factors [89]. Mitochondria also participate in systemic signaling to coordinate organismal stress responses and mediate systemic metabolic changes, as well as function as hubs of immune signaling [89].

Integrated interoceptive and exteroceptive control of organismal energy metabolism allows for fine adjustment of energy metabolism to internal and external environment [90]. The living organism is capable of signaling its activity and the signals are sent to all organs and receive feedback about the body’s needs. The exchange and processing the sensory information in the brain plays a fundamental role. Internal information metabolism is based on the subconscious exchange of cellular and hormonal signaling, which are the base of subjective motivational and emotional drive [73]. Additionally, the exchange of information with the environment allows us to create in the brain a sphere of self-consciousness and egocentric representation of our environment [86].

### 4.3. Role of the Kynurenine Pathway

Contributing to glycan–protein interactions, tryptophan is indispensable to sustaining cellular life. Up to 90% of the tryptophan is catabolized through the kynurenine pathway, which produces various metabolites, with kynurenic and quinolinic acids as the most important intermediates [91]. In the brain, kynurenic acid acts as a glutamate (NMDA) receptor and Ca^2+^ ion channel antagonist, whereas quinolinic acid plays the opposite function [91]. The glutamate-dependent opening of the Ca^2+^ channel allows calcium to enter the cytoplasm, and then the channels are blocked by Mg^2+^ ions, allowing neurons to recover their resting potential. Thus, the quinolinic acid may disturb the physiological functioning of neurons, causing calcium overload and cell death [91]. About 1% of dietary tryptophan is involved in serotonin synthesis in the brain, regulating diverse processes including mood, cognition, reward, learning, and memory. Melatonin is another metabolite produced in the serotonin pathway that influences the digestive, immune, and reproductive systems and diurnal rhythms.

In the brain, most of the tryptophan is metabolized into 5-hydroxytryptamine, leading to lower concentrations of quinolinic acid [91]. The quinolinic acid concentration increases during immune responses. During inflammation, infiltrating macrophages, dendritic cells, and microglia are the main sources of excessive quinolinic acid concentration in the brain. Only at low concentrations, quinolinic acid is catabolized to NAD^+^ to fulfill a neuroprotective role. Excess of quinolinic acid causes saturation of the catabolic system, resulting in neurotoxic effects [91].

Signaling related to the efficiency of energy metabolism is crucial for life. The decrease in NAD^+^ levels and the possibility of restoring its physiological level play an important role in the control of motivational and emotional mechanisms that are the basis of information metabolism. In a young, healthy organism, tryptophan catabolism via the kynurenine pathway is the primary source of NAD^+^ supplementation. Tryptophan in the blood activates the brain’s serotonergic reward system [92]. Additionally, in the ventromedial hypothalamic nuclei, serotonin signals recruit two mediators to inhibit bone mass increase, the neuropeptide cocaine and amphetamine-regulated transcript (CART) and adrenaline [70]. However, high-stress levels or chronic infections can change this process by provoking an immune response that alters the kynurenine pathway. Instead of serotonin in the intermediate steps, several bioactive compounds are produced, such as kynurenic acid and quinolinic acid [92]. These intermediate metabolites have an immunomodulatory effect depending on coexisting inflammatory processes. For example, kynurenine inhibits the synthesis of deoxyribonucleic acid, leading to necrosis and tumorigenesis [92]. In turn, quinolinic acid produced in the brain by microglia excessively stimulates NMDA receptors that may culminate in excitotoxicity and death of overactive dopaminergic neurons of the substantia nigra and striatum. Quinolinic acid is also associated with kidney and liver failure, and neurodegenerative conditions, leading to depression, anxiety, sleep problems, and cognitive changes [92]. Quinolinic neurotoxicity initiates a cascade of events culminating in excitotoxicity, ATP depletion, oxidative stress, neuroinflammation, and selective GABAergic neuron loss [93]. Importantly, the quinolinic acid neurotoxicity can be alleviated by melatonin, which acts independently and by different mechanisms in modulating antioxidant enzyme activities [94].

## 5. Signaling of the Motor Activity

Motor activity is the main output of the nervous system. It allows the conversion of a substantial amount of biochemical energy into mechanical work. To realize this process effectively, both muscles and bones must precisely signal their activity. Skeletal muscle is an endocrine organ, which secretes hundreds of myokines that exert their effects in autocrine, paracrine, or endocrine manners [95]. Myokines allow crosstalk between the muscle and other organs, including the brain, adipose tissue, bone, liver, gut, pancreas, and vascular bed [95]. Myokines are responsible mainly for mediating energy supply, muscle cell proliferation, differentiation, and regeneration. Myokine IL-6 affects lipid and glucose metabolism and plays important roles in myogenesis. IL-6 signaling within the muscle can affect glucose uptake and fat oxidation via AMPK activation. Musclin has been identified as an exercise-induced factor promoting skeletal muscle mitochondrial biogenesis. Myostatin is a positive regulator of bone resorption [95]. BAIBA is a molecule produced by contracting muscles [54]. It activates the β-oxidation pathway of hepatic fatty acid and improves insulin sensitivity in skeletal muscle. It also protects osteocytes against reactive oxygen species and prevents bone and muscle loss [54]. The function of BAIBA is lost in aging bones due to the downregulation of its receptor in osteocytes [54].

The energetic processes occurring in the neuromuscular system depend on the intra- and extracellular Ca^2+^ signaling [96,97]. Serum calcium levels are maintained within the physiological range by bone resorption [98]. Calcitonin is secreted by the para-follicular cells of the thyroid gland in response to an increase in serum calcium concentration opposing the effects of parathyroid hormone [99,100]. In bones, calcitonin inhibits osteoclast action and bone resorption [96,101]. Calcitonin in kidneys reduces the reabsorption of calcium, sodium, potassium, chloride, and phosphate. Respiratory alkalosis decreases the serum ionized calcium, whereas metabolic acidosis is associated with an increase in calcium excretion, independent of parathormone changes [98]. The activity of calcitonin in the central nervous system may induce eating disorders [100].

Another mediator of body–environment interaction is vitamin D. It is a hormone that plays a major role in the regulation of calcium and phosphate metabolism as well as promoting bone health and bone remodeling. Vitamin D is the essential catalyst for energy homeostasis and glucose metabolism, influencing insulin secretion and glucose levels. It is synthesized from its cholesterol precursor, which can be converted in the skin to the active form of the vitamin after sunlight exposure [102]. Cholesterol and vitamin D are essential for life since their activity is required to build and maintain cellular and mitochondrial phospholipid membranes. Vitamin D increases the activity of the tyrosine hydroxylase in adrenal medullary cells, affecting the synthesis of neurotrophic factors, nitric oxide synthase, and glutathione, which control the body’s response and adaptation to stress [103]. The half-life of vitamin D is 8–10 h, and its levels are affected by changes in calcium, phosphorus, parathyroid hormone, and fibroblast growth factor 23 (FGF 23) levels [102]. It is estimated that about 3% of the human genome is regulated by vitamin D, which controls two major cellular functions, proliferation and differentiation [102].

Calcium ions play also a vital role in motor activity. In hypoactivity, calcium excretion is increased while absorption is reduced leading to a sustained negative calcium balance. Osteoblasts and osteoclasts consume up to 20% of the quantity of glucose taken up by muscles [70]. Inside the osteoblast, glucose is metabolized mostly through aerobic glycolysis to generate ATP molecules necessary for bone formation [70]. Insulin signaling in osteoblasts favors osteoclast differentiation and bone resorption by inhibiting the expression of osteoprotegerin [70]. Insulin and IGF-1 signaling in osteoblasts increases the release of osteocalcin, the hormone protein that stimulates insulin secretion [70]. Osteocalcin in contracting muscles intensifies the use of the self-stored glycogen and consequently enables postprandial insulin-dependent glucose uptake [104]. This results in a decline in muscle mass and strength, which limits physical activity. Skeletal tissue is a major storage site for calcium and phosphate ions, and endocrine organs secrete peptides working on other remote organs [105]. Bones secrete osteocalcin, a hormone that participates in glucose and fat metabolism [52]. Osteocalcin also stimulates insulin secretion by the pancreas as well as β-cell proliferation.

Bones communicate with bone marrow, muscle, adipose tissue, kidney, liver, and brain [105]. The bone-derived hormone osteocalcin is the regulator of energy metabolism [76,105,106]. Osteocalcin intensifies the release of insulin in the pancreas with concomitant activation of adiponectin in the adipocytes. This amplifies insulin sensitivity in skeletal muscles, fat, and hepatic tissue [44]. In physiological conditions, the plasma levels of osteocalcin and glucose are negatively correlated, while hyperglycemia induces a low bone turnover rate by evoking osteoblast dysfunction and suppressing serum osteocalcin levels [107].

In fat cells, osteocalcin triggers the release of the hormone adiponectin, which increases insulin sensitivity [44,108]. Adipokines produced by fat cells, such as leptin and adiponectin, are key mediators of physiological processes in distant organs, such as the brain, liver, and muscle, where they control appetite, digestion of nutrients, energy expenditure and storage, glucose and lipid metabolism, and insulin sensitivity.

The regulation of bone remodeling by an adipocyte-derived hormone implies that bone may exert a feedback control of energy homeostasis. Osteocalcin signaling in myofibers is necessary for adaptation to exercise by favoring the uptake and catabolism of glucose and fatty acids [104]. Circulating levels of osteocalcin double during aerobic exercise at the time those of insulin decrease [104]. In humans circulating levels of osteocalcin decrease during early adulthood but the supplementation of osteocalcin can restore exercise capacity and, thus, can reverse its age-induced decline [104].

Osteocytes control bone resorption through the osteoclast-released receptor activator of nuclear factor kappa-Β ligand (RANKL), bone formation through the local release of sclerostin, which inhibits Wnt/β/catenin, and phosphate metabolism through the systemic production of FGF-23. Prostaglandin (PGE2) and nitrogen released by osteocytes have anabolic effects on osteoblasts. Osteocytes are exposed to hormones, cytokines, inflammatory factors, and signals from other tissues and glands, such as muscles, kidneys, intestines, and parathyroid glands, which regulate the activity of osteocytes [109]. Metabolic activity and bone remodeling are controlled by mechanical factors, endocrine and paracrine signals including insulin-like growth factor, transforming growth factor TGF-β, interleukins IL-1, IL-6, and tumor necrosis factor-α (TNF-α) [109].

Osteopontin is an extracellular structural protein that plays a critical role in bone synthesis mediated by osteoblast activity. The inorganic phosphate is a potent inhibitor of osteopontin which during intensive motor activity inhibits the deposition of hydroxyapatite in bones, increasing calcium ions in circulation [32]. The calcium ions bind to osteocalcin which transports them to the heart, diaphragm, striatal muscles, and brain. In these tissues, using calcium-induced calcium signaling enables neuromuscular activity [32].

The adipose tissue secretes several adipokines such as leptin, adiponectin, adipsin, resistin, visfatin, and lipokines regulating systemic glucose and lipid metabolism [110,111,112]. Leptin, the satiety hormone secreted by adipocytes in the presence of insulin, prevents overnutrition by inhibiting the enzyme, 5′-adenosine monophosphate-activated protein kinase (AMPK) in the hypothalamus to suppress appetite [113]. Simultaneously, it activates AMPK in skeletal muscles, by increasing the AMP/ATP ratio, which stimulates glycolysis and glycogen synthesis [44]. AMPK regulates mitochondrial biogenesis by regulating PGC1α, a cofactor that promotes the transcription of nuclear-encoded mitochondrial genes [113]. The accumulation of three major nutrients, glucose, fatty acids, and amino acids is suggested to suppress AMPK and contribute to insulin resistance [113].

Ghrelin is a hormone primarily produced by enteroendocrine cells of the gastrointestinal tract, especially the stomach, and increases the drive to eat. Ghrelin prepares organisms for food intake by increasing gastric motility and stimulating the secretion of gastric acid. Blood levels of ghrelin are highest when hungry, returning to lower levels after meals. Marrow adipose tissue secretes adiponectin which exerts systemic metabolic effects by regulating glucose levels and fatty acid breakdown [114]. Low levels of adiponectin are symptomatic of mitochondrial dysfunctions related to inflammation, hypoxia, or endoplasmic reticulum stress [114].

The liver-released fetuin-A participates in a glucose and lipid metabolic switch [115]. Fetuin-A inhibits insulin receptors in the liver, skeletal muscles, and fat tissue, thus reducing glucose uptake and promoting lipid-induced insulin resistance [115,116]. It also acts as an inhibitor of bone calcification. Free fatty acids cause fetuin-A overexpression by increasing the proinflammatory protein NF-κB [116].

## 6. Senescence and Death

Age-related decline in metabolic functions, their efficiency, and control ability across multiple physiologic systems compromises the ability of the organism to control allostasis, resulting in the escalation of systemic involution. It is characterized by the gradual decline in life-sustaining processes at the micro and macro levels. At a cellular level, involution is characterized by energy dysmetabolism, leading to apoptosis. Deficiency in cellular repair processes and turnover impairs the proper functioning of individual organs, leading to frailty. Frailty is an extreme consequence of aging, where the metabolic deficiency accelerates and homeostatic responses begin to fail [117]. Terminal frailty syndrome is associated with the escalation of the chronic inflammatory process that accompanies aging, negatively impacting systemic energy metabolism. The process strikes the cardiovascular, endocrine, musculoskeletal, and respiratory systems. The observed rapid reduction in muscle mass impairs life-defining activities, culminating in cardiovascular and metabolic disorders [117]. The most striking effect of systemic involution is neurodegeneration, depriving the organism of the axial control of all life (Figure 2).

The process of aging points to a series of micro-changes at the cellular and tissue levels that culminate in irreversible pathological changes in the organismal macrostructure [118]. Cellular senescence occurs in response to endogenous and exogenous stresses, including telomere dysfunction, oncogene activation, and persistent DNA damage [118]. Senescent cell extrinsic activities, broadly related to the activation of a senescence-associated secretory phenotype, amplify the impact of cell-intrinsic proliferative arrest and contribute to impaired tissue regeneration, chronic age-associated diseases, and organismal aging [118].

Progressive system senescence and metabolic dysregulation interfere with the basic life processes. Limited longevity of individuals opens the need for species life based on reproduction. It allows descendants to better adapt to the slowly but continually changing environment. A characteristic feature of ontogeny is the change in the adaptative processes. The most intensive adaptation to the environment is observed in the early developmental period, from birth to maturity. Then the adaptive abilities of the organism gradually weaken with age, which results in a decrease in the body’s immunity, and chronic low-grade inflammation. Particularly, dysfunctions of cellular turnover and defective repair processes increase inflammatory processes. Additionally, thymus involution and the resulting failure of the acquired immune system cause exacerbation and chronicity of inflammation. Intracellular protein accumulation, typical of aging highly active cells, reduces the oxygen diffusion coefficient, intensifying the energy crisis. Senescence is a cellular response featuring a stable cell cycle arrest that limits the potential of cell proliferation [119]. The senescent mesenchymal stem cells are characterized by decreased stemness, cell phenotype changes, immunomodulatory property damage, impaired proliferative ability, and higher susceptibility to apoptosis, which highly restricts their therapeutic value [119].

The senescence of the bone marrow strikes particularly the process of hematopoiesis. Erythrocytes, under physiological conditions, are replaced on average every 120 days [50], their lifespan in older organisms is shortened to up to 14 days [120]. As a consequence, intense but not fully efficient hematopoiesis is forced, resulting in the production of dysmorphic red blood cells, which not only causes their rapid use but also intensifies damage to blood vessels. Transport and distribution of energy substrates such as oxygen, glucose, calcium, and phosphates decreases rapidly, causing the development of metabolic dysfunctions. The main of these is the escalating shift in energy metabolism to beta-oxidation.

The thymus involution is an example of a life-limiting process, aiming at dysregulations of immunoreactivity. The thymus is a specialized primary lymphoid organ of the immune system that controls immuno-adaptive processes [121,122,123]. Hemopoietic stem cells of bone marrow are the source of T-cell precursors, which are then transported by blood to the thymus, where they differentiate and mature. Most of the thermus activity falls during the early developmental period and later on, due to natural involution. Size and activity of the thymus decline steadily, and its cells are gradually replaced by fatty tissue, which exhausts the body’s adaptive abilities of the organism, culminating in death [122].

Human thymic involution starts as early as 1 year of age [122]. At the age of 45 years, adipose tissue constitutes almost 75% of the thymus volume, morphologically consisting of multiple lipid-laden multilocular cells with proinflammatory properties [123]. Consequently, thymic involution is associated with increased susceptibility to many diseases, including cancer, infection, and autoimmunity [122]. In the elderly, nearly all the thymus parenchyma consists of adipocytes. The resultant lymphocyte T deficiency directly impacts the development of inflammation and induces various autoinflammatory disorders, including atherosclerosis [123].

The primary issue in energy metabolism is insulin resistance. It is a disorder with a multifactorial basis directly related to glucose metabolism or excessive fat accumulation in the body. Insulin has a substantial impact on intestinal lipid metabolism both directly and indirectly by inhibiting the release of fatty acids from adipose tissue. In adipocytes, insulin stimulates glucose uptake to fuel de novo lipogenesis [124]. Additionally, insulin reduces the formation of intestinal lipoproteins. During passage through the hepatic sinusoids, remnant particles are further hydrolyzed by hepatic triglyceride lipase and gain additional apoE, which makes it possible for them to bind to and be taken up by proteins on the surface of liver cells [125]. The accumulation of the remnant lipoproteins plays a pivotal role in fatty liver disease.

The main pathogenic impact of lipid-based energy metabolism is visceral fat accumulation, leading to lipotoxicity [126]. Fat metabolism is combined inextricably with visceral fat accumulation and impairs metabolism in all tissues that become resistant to the anti-lipolytic effect of insulin [126]. With the decline of insulin anti-lipolytic action, the intracellular levels of free fatty acid increase rapidly, leading to endoplasmic reticulum stress. Moreover, the excess of free fatty acids induces chronic inflammation that is harmful to multiple organs and systems [126]. Dominance of fat metabolism may induce hepatic steatosis and insulin resistance, concurrent with innate immune system activation [127].

Due to the inherently limited efficiency of biological processes, the basis of life is multimodal interaction with the environment, the purpose of which is the precise and efficient adaptation of the organism to its natural environment. The process of adaptive adjustment also applies to building immune protective barriers. Cellular senescence can contribute to systemic aging and all age-related pathologies by accelerating the loss of tissue regeneration through the depletion of stem cells and progenitor cells [61].

## 7. Conclusions and Directions for Future Research

An important but not yet fully understood process that determines cell life is the mitochondrial–lysosomal interaction. This highly dynamic process involves rapid transformation of the mitochondrial structure, depending on the availability of energy metabolites and the level of cellular activity. Through mitochondrial fusion and fission, the process of cellular energy metabolism is continuously optimized [128]. During these processes, lipoprotein structures are broken down and then used to rebuild mitochondria. However, in long-lived cells, such as neurons, cardiomyocytes, and osteocytes, there is an accumulation of waste products that disrupt this essential process for life. In particular, the accumulation of lipofuscin in lysosomes, which cells are unable to remove or break down, causes irreversible damage to the vital processes of long-lived cells. Therefore, understanding this process and the elimination of lipofuscin from aging cells is an important challenge for future research.

The next challenge for contemporary science is research on the contribution of the kynurenine pathway to the progression of senescence. The end products of this pathway are several neurotransmitters that establish mood, in addition to supplementation of the aging-related NAD^+^ deficit. The latter function in the elderly is completely ineffective due to multiple inflammatory processes associated with senescence, causing the brain to respond with anxiety and stress. Although several NAD supplements are currently available, none of them correct the malfunctioning of the entire kynurenine pathway.

Finally, there is an intriguing mystery in human cellular structure, which solving may help to slow down senescence. In all mammals except humans, the cellular glycocalyx is composed of N-glycolylneuraminic acid (Neu5Gc). Human cells instead use Neu5Ac for this purpose. Essential resources of this sialic acid are obtained body during pregnancy and then form breast milk feeding only. Later in life, Neu5Ac resources are not replenished, and consequently, depletion leads to breakdown of synaptic processes in the brain related to learning and memory.

In other cells, replacing Neu5Ac with dietary Neu5Gc disrupts cellular metabolism, leading to changes in cellular signaling, immunoreactivity, energy metabolism, and promoting carcinogenesis. Therefore, developing a method to supplement this sialic acid in adults seems to be a promising method of slowing down the senescence and recovery of brain function. None of the proposed solutions here can completely inhibit the aging process, but they can slow it down and thus improve the quality of life of the elderly.

## Figures and Tables

**Figure 1 metabolites-15-00404-f001:**
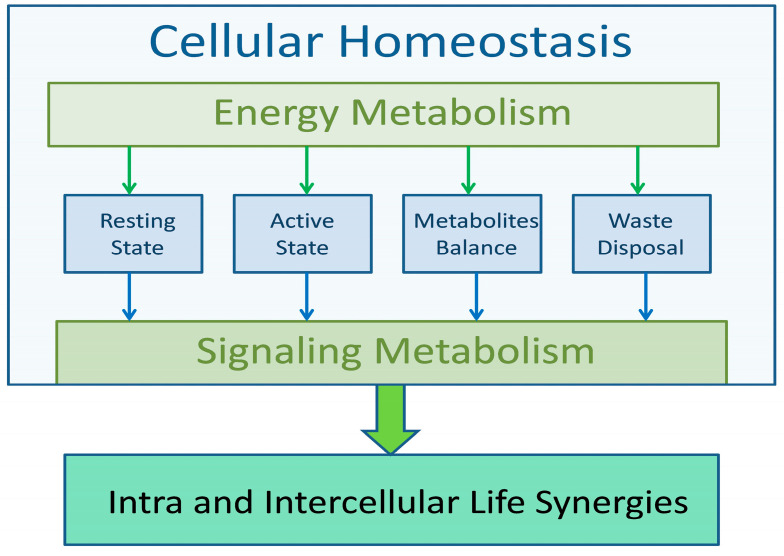
The diagram shows the algorithm of life processes at the cellular level. The condition for the correct implementation of life processes is the supply of energy substrates to cells embedded in a stable extracellular environment and maintaining homeostasis. Maintaining homeostasis means precise control of physicochemical parameters, including pH, temperature, and correct concentrations of substrates necessary for the implementation of cellular respiration. Under the influence of external signals, so-called active cells can change their energy state and perform their life functions. Worn and dysfunctional cells are replaced in the cellular turnover process. Only single-living cells, especially neurons and cardiomyocytes, are not subject to replacement and their lifespan determines the life of the organism.

**Figure 2 metabolites-15-00404-f002:**
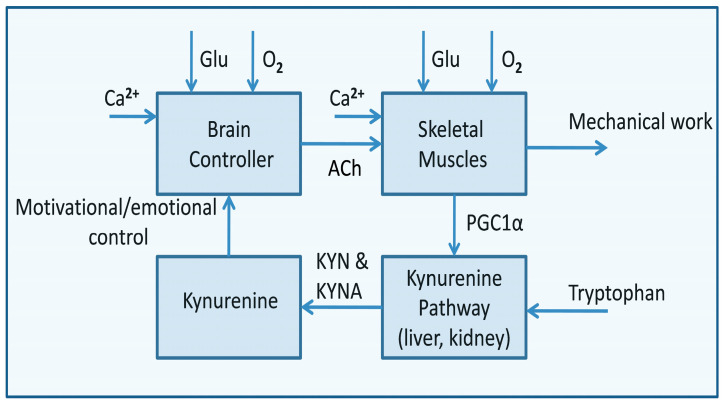
Schematic diagram showing intertissue interaction during motor activity. Neuronal activity and release of acetylcholine (ACh) at the neuromuscular junction depend on calcium signaling, which activates the calcium-triggered calcium release. It modulates the interaction between lysosomes and mitochondria, dynamically adjusting the morphological structure of both organelles. In this process, the muscles release PGC1alfa, which in turn tunes the activity of the kynurenine pathway. The pathway accounts for 90% of peripheral tryptophan metabolism. One of these metabolites, kynurenine (KYN) can cross the blood–brain barrier, modulating glutamine transmission and plasticity in the central motor area. Consequently, both KYN and kynurenic acid (KYNA) may contribute to depression, neuronal cell death, and neuroinflammation.

## Data Availability

No new data were created or analyzed in this study. Data sharing is not applicable to this article.

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
