# Peer review of "Biomolecular Basis of Life"

_metabolites, 2025, doi:10.3390/metabo15060404_

Round 1
Reviewer 1 Report
Comments and Suggestions for Authors
Comments to the Authors
The manuscript of Janusz WiesÅ‚aw BÅ‚aszczyk, “Biomolecular Basis of Life” is a review that presents a broad, ambitious narrative article aiming to systematize fundamental biological processes that underpin life, particularly focusing on energy exchange and metabolism. It attempts to analyze a wide range of physiological, biochemical, and systemic aspects of life into a coherent theoretical framework. The review needs a thorough revision in order to capture the interest of a wide audience, specifically, a drastic revision of the structure of the manuscript and more practical information on the approach would be needed. The perspectives would gain in interest if the ideas developed were more structured and mature. Hopefully, my remarks will help the authors gain clarity and interest. This manuscript presents a potentially valuable and original analysis but requires substantial reorganization, clarification, and scholarly support before it can be suitable for publication.
- The manuscript has many repetitions and duplicates: a) on word level where the word “environment” is used three times: “To survive in a changing environment, the body must maintain homeostasis, i.e., a stable internal environment despite changes in the environment” (Lines 32-32); b) on sentence level “Catecholamines are the primary activators of lipolysis.” (Lines 149 and 158); c) on paragraph level with whole copy-pasted paragraphs (Lines 56-65, 773-782);
- The manuscript lacks clear sectioning and transitions between topics. For example, it jumps from molecular mechanisms to social behavior and brain consciousness without explicitly delineating the conceptual bridge. Consider choosing a more specified topic for the review.
- Statements containing the word “self-awareness” (Lines 15, 809) are intriguing but speculative without empirical backing. Such statements should be clearly marked as hypotheses or philosophical interpretations, not established facts.
- The abstract is too general and does not accurately reflect the depth and focus of the full manuscript. Revise the abstract to include key concepts, mechanisms discussed, and the main thesis or contributions of the review.
- Figures, tables, or diagrams could significantly enhance reader comprehension, especially for complex signaling or metabolic pathways.
- kinematics --> kinetics (Lines 315, 783). Kinematics is related to the study of locomotion, the movement of organisms in biology, or the moving parts of an engine in engineering. Kinetics is related to the rate of chemical reaction.
- Several aspects, such as novel strategies and analytical platforms for study of metabolites (GC, HPLC, CE coupled to MS and NMR spectroscopy)1, bioinformatics pipelines and databases (MetaboLights)2 are unfortunately neglected.
REFERENCES
- Kuehnbaum, N. L.; Britz-McKibbin, P., New advances in separation science for metabolomics: resolving chemical diversity in a post-genomic era. Chem Rev 2013, 113 (4), 2437-68.
- Yurekten, O.; Payne, T.; Tejera, N.; Amaladoss, F. X.; Martin, C.; Williams, M.; O'Donovan, C., MetaboLights: open data repository for metabolomics. Nucleic Acids Res 2024, 52 (D1), D640-D646.
Author Response
Dear Reviewer. Thank you very much for your comments and advice, which helped me improve my manuscript. I have re-edited the manuscript according to your advice.
In particular:
Ad. 1. The repetitions and duplicates have been removed.
Ad. 2. The sectioning and transitions between topics have been corrected. I have removed the redundant texts and added new links to the discussed phenomena. The added text is marked in red font.
Ad. 3. I am convinced that self-awareness is the culmination of the information metabolism; thus, I would like to leave this fragment as is.
Ad. 4. I corrected the abstract slightly, but to make it less general in 200 words is over my mental capability.
Ad. 5. I corrected the abstract slightly, but to make it less general in 200 words is over my mental capability.
Ad. 6. I am sorry for this stupid mistake. Of course, I have corrected to "kinetics". Thank you.
Ad. 7. Unfortunately, I am no longer able to add the topics you suggested. I will try to do so in the next article, or encourage others to do so.
Reviewer 2 Report
Comments and Suggestions for Authors
The review article titled as “Biomolecular Basis of Life” the authors address a fascinating and ambitious topic with potential to contribute to scientific discourse. However, its current lack of coherence, oversimplified claims, and vague assertions prevent it from meeting the standards of a top-tier journal. With significant restructuring, clearer definitions, and better-supported arguments, this review could become a compelling synthesis of life phenomena. I encourage you to revise and resubmit, focusing on the suggestions provided.
Comments
-The scope of the review (e.g., human-specific vs. all biological systems) is ambiguous, making it hard to assess its focus. The goal to "systemize our knowledge" and "initiate a broader discussion" is vague—what new insights or perspectives does this review offer?
-The "brain representation of our body and the egocentric representation of the environment" feels tacked on, with no clear tie to metabolism or signaling.
-The "life/death cycle" at cellular and social levels is an interesting concept, but its relevance to metabolism or the review’s theme is unclear.
-The link between cellular communication, self-awareness, and "conscience social interactions" is intriguing but lacks specificity or evidence. How does signaling metabolism lead to self-awareness? This is a significant leap requiring justification.
-The claim "All energy-dependent life processes are controlled by phosphate and calcium signaling" is overly absolute. While these are critical, other signaling pathways (e.g., cyclic AMP, MAPK) are equally important.
Good luck…
Author Response
Question #1: What new insights or perspectives does this review offer?
Response: For several years now, I have been observing the scientific discussion on explaining the phenomenon of life. This is an extremely difficult challenge because each of us, depending on our knowledge and scientific interests, looks at and evaluates this phenomenon differently. That is why I wrote this narrative review article to provoke an exchange of knowledge about life. In this article, I wanted to present an interesting hypothesis that there are many analogies between life at the cellular, organismal and even ecosystem level. In particular, I wanted to show that all forms of life are based on the synergy of energy and information metabolism. I realize that we are still far from explaining the phenomenon of life, but finally a broad discussion on this topic should begin, and this is the goal of my review.
Criticism #2 The brain representation of our body and the egocentric representation of the environment feels tacked on, with no clear tie to metabolism or signaling.
Response: It depends on how we look at the phenomenon of life. The brain as a controller of life has a specific goal of realizing this control. Therefore, organs and cell groups signal their life needs to the brain. This is why we have a feeling of hunger or satiety. After all, it is the brain that directs us towards food sources (food).
Criticism #3: The "life/death cycle" at cellular and social levels is an interesting concept, but its relevance to metabolism or the review’s theme is unclear. Response: The basis of life is energy and information metabolism. Both are based on the exchange of energy and signals with the environment, allowing for precise adaptation of the organism to the environment. Such adaptation depends on the optimal structure and function of the organism, assuming at the same time that our environment does not change. Unfortunately, this is not the case, and that is why the process of evolution has chosen the mechanism of exchange of cells and individuals and replacing them with new ones, better adapted to the environment.
Question 3: How does signaling metabolism lead to self-awareness? This is a significant leap requiring justification.
Response: The brain encodes the goals and mechanisms for fulfilling our life needs. For example, defensive reflexes allow a person to save their life in case of danger. The same applies to food behaviors and adapting them to the needs of our body. That is why we eat meals to maintain constant levels of energy fuels, or enter into social interactions, e.g. by getting married, despite the fact that from the point of view of energy metabolism it makes no sense.
Question 4: The claim "All energy-dependent life processes are controlled by phosphate and calcium signaling" is overly absolute. While these are critical, other signaling pathways (e.g., cyclic AMP, MAPK) are equally important.
Response: I have clarified these issues in the revised version of my article. Thank you for bringing this to my attention.
Reviewer 3 Report
Comments and Suggestions for Authors
The manuscript is an ambitious narrative review exploring energy and information metabolism, covering glucose/lipid metabolism, calcium/phosphate signaling, cellular turnover, senescence, and their roles in life and aging. Its interdisciplinary scope, linking molecular, cellular, and systemic perspectives, is commendable. However, the manuscript lacks clarity, coherence, and scientific rigor due to its broad scope, unsupported claims, and verbose writing. Major revisions are needed to enhance focus, accuracy, and readability.
- Lack of Focus and Structure: The manuscript covers too many topics (e.g., ATP signaling, brain function, social interactions) without a clear narrative thread. Sections are poorly organized, with abrupt transitions.
- Scientific Inaccuracies: Unsupported claims (e.g., signaling metabolism leading to self-awareness) and errors (e.g., glycolysis yielding 32 ATP) undermine credibility.
- Inconsistent Referencing: Citations are uneven, with some sections lacking references and others relying on outdated sources.
- Verbose Writing: Repetitive phrasing and grammatical errors reduce readability. Technical terms are used inconsistently or undefined.
- Speculative Tone: Broad, unsupported statements (e.g., chronic energy deficit as the greatest threat) lack nuance.
- No Visual Aids: Complex topics like metabolic pathways need diagrams for clarity.
Author Response
Criticism #1. Lack of Focus and Structure: The manuscript covers too many topics (e.g., ATP signaling, brain function, social interactions) without a clear narrative thread. Sections are poorly organized, with abrupt transitions.
Response: In the revised version of the article, I have tried to correct these errors. I have removed some of the redundant text and re-divided the individual sections so that each of them has a specific purpose.
Criticism # 2. Scientific Inaccuracies: Unsupported claims (e.g., signaling metabolism leading to self-awareness) and errors (e.g., glycolysis yielding 32 ATP) undermine credibility.
Response: I treat my article as a form of discussion with readers, and therefore, I have presented some threads that are causing discussion. I hope that for this narrative review, I will also meet with criticism and competent adversaries will explain the problem of self-awareness. As for these 32 ATP molecules, this is not my invention but a literal quote from the publication [2], which is a publicly available source of scientific knowledge for students.
Criticism #3: Inconsistent Referencing: Citations are uneven, with some sections lacking references and others relying on outdated sources.
Response: I apologize, but I chose citations appropriate to my views and scientific knowledge. It is very difficult to keep up with the avalanche of new publications these days. Moreover, in my article, places where there are no references mean that the knowledge is textbook or these are my personal views, which I would like to mobilize readers to make critical remarks or alternative hypotheses.
Criticism # 4. Verbose Writing: Repetitive phrasing and grammatical errors reduce readability. Technical terms are used inconsistently or undefined.
Response: I am very sorry. In the revised version, I have improved the text and removed redundant fragments. Unfortunately, I am not a native speaker and my pEnglish may raise many critical remarks. Additionally, I use knowledge and publications from various fields of science: from biology and medicine to biocybernetics and biomedical engineering. Each of these fields has its own scientific jargon, which may irritate the reader. Therefore, I apologize. For me, the most important issue is explaining the phenomenon of life despite my writing problems.
Criticism # 5. Speculative Tone: Broad, unsupported statements (e.g., chronic energy deficit as the greatest threat) lack nuance.
Response: In one of my scientific incarnations, I was a neurophysiologist and behaviorist, so I don't focus on statements that may be obvious. Undernourished animals don't reproduce and eventually die. To me, this is pure energy deficit.
Criticism # 6. No Visual Aids: Complex topics like metabolic pathways need diagrams for clarity.
Response: Thanks for the advice. In the revised version of the article, I added, at your suggestion, Figure 1, which shows my view of the organization of life processes at the cellular level.
Round 2
Reviewer 1 Report
Comments and Suggestions for Authors
Despite all corrections, the review is still general and superficial. How does this manuscript differ from many other reviews devoted to studying metabolites in living organisms?
- Throughout the text of the manuscript: NAD ---> NAD+
- Line 182: The authors mention sirtuins and PARPs as a NAD+-consuming enzymes but omit the third class of enzymes. "At the cellular level, NAD is continually turned over by THREE classes of NAD-consuming enzymes: the NADases, the protein deacetylase family of sirtuins, and PARPs, which have various important cellular functions impacting metabolism, genomic stability, gene expression, inflammation, circadian rhythm, and stress resistance [12,13,14,15]"
- The review manuscript lacks a well-structured and concise conclusion. For the integrity of the manuscript, it is necessary to include conclusions with a brief description of current achievements in the study of metabolic pathways in living organisms, as well as promising directions for future research.
The English could be improved to more clearly express the research.
Author Response
- Throughout the text of the manuscript: NAD ---> NAD+. Response: It has been corrected in the present version.
- Comment: Line 182: The authors mention sirtuins and PARPs as NAD+-consuming enzymes but omit the third class of enzymes. "At the cellular level, NAD is continually turned over by THREE classes of NAD-consuming enzymes: the NADases, the protein deacetylase family of sirtuins, and PARPs, which have various important cellular functions impacting metabolism, genomic stability, gene expression, inflammation, circadian rhythm, and stress resistance [12,13,14,15]" Author response: I have added the third class of NAD-consuming enzymes i.e., glycohydrolase/CD38.
- The review manuscript lacks a well-structured and concise conclusion. For the integrity of the manuscript, it is necessary to include conclusions with a brief description of current achievements in the study of metabolic pathways in living organisms, as well as promising directions for future research. Author response: I have restructured my manuscript slightly by adding "Conclusion and directions for future research."
Reviewer 2 Report
Comments and Suggestions for Authors
- Question #1: The explanation is partially correct but lacks specificity about the review’s novel insights. It’s more aspirational than concrete.
- Criticism #2: The explanation is incorrect or incomplete, as it does not adequately link metabolism to brain representations beyond basic signaling for hunger.
- Criticism #3: The explanation is conceptually sound but lacks detail and evidence to fully connect the life/death cycle to metabolism.
- Question #3: The explanation is scientifically incorrect, as it fails to address the mechanistic basis of self-awareness.
Given the responses, the manuscript does not yet meet the standards for acceptance but shows potential if the authors can address these gaps with greater scientific rigor and clarity. I encourage resubmission after thorough revision.
Author Response
- Question #1: The explanation is partially correct but lacks specificity about the review’s novel insights. It’s more aspirational than concrete. Response: Response: I am sorry, but this review reflects only my subjective point of view, and without a clear comment, I cannot correct it to satisfy you.
- Criticism #2: The explanation is incorrect or incomplete, as it does not adequately link metabolism to brain representations beyond basic signaling for hunger. Response: I cannot agree with your criticism. In the present version, I have provided a wide explanation of nervous control.
- Criticism #3: The explanation is conceptually sound but lacks detail and evidence to fully connect the life/death cycle to metabolism. Response: I had rather limited space for this subject, and I encourage you to write something on this matter.
- Question #3: The explanation is scientifically incorrect, as it fails to address the mechanistic basis of self-awareness. Response: Response: I am only a simple scientist, and therefore I cannot criticise or omit to mention self-awareness. This is both a philosophical and a medical conception of self-awareness, which deserves a separate review.
Reviewer 3 Report
Comments and Suggestions for Authors I want to see the second revised version for Improvement 1. Provide more detailed mechanistic insights into critical processes like the kynurenine pathway, sialic acid metabolism, or mitochondrial dysfunction. 2. Streamline transitions between sections to create a more unified narrative such as metabolic signaling, cellular maintenance to improve cohesion. 3. Discuss potential therapeutic interventions (e.g., NAD boosters) to bridge basic science with clinical applications. 4. Additional diagrams illustrating metabolic pathways or inter-organ crosstalk for enhancing clarity.Author Response
The final version is included. I corrected the present version according to the Reviewer's suggestions. In particular, I have explained the role of the kynurenine pathway in senescence. Additionally, in ch.7 I provided my subjective point of view on future directions of research and potential therapeutic strategies in aging, focusing on NAD, sialic acid, Neu5Ac, and lysosome-mitochondrial interaction with lipofuscin as the main toxic metabolite.
Round 3
Reviewer 1 Report
Comments and Suggestions for Authors
The author has revised the manuscript, so there are no more comments or questions.
Author Response
I have added the second figure explaining the signalling metabolism during movement.
Reviewer 2 Report
Comments and Suggestions for Authors
I am satisfied that the author has provided very sutibale responses. I appreciate their acadamic and thoughtful responses. Good luck.
Author Response
Please find the final version of my narrative review. Thank you very much for your healp.